# Parental Pesticide Exposure and Childhood Brain Cancer: A Systematic Review and Meta-Analysis Confirming the IARC/WHO Monographs on Some Organophosphate Insecticides and Herbicides

**DOI:** 10.3390/children8121096

**Published:** 2021-11-28

**Authors:** Joseph Feulefack, Aiza Khan, Francesco Forastiere, Consolato M. Sergi

**Affiliations:** 1National “111” Center for Cellular Regulation and Molecular Pharmaceutics, Key Laboratory of Fermentation Engineering (Ministry of Education), Hubei University of Technology, Wuhan 430068, China; feulefac@ualberta.ca; 2Department of Lab. Medicine and Pathology, University of Alberta, Edmonton, AB T6G 2R3, Canada; draizakhan@gmail.com; 3Department of Epidemiology, Regional Health Service of Lazio, 00147 Rome, Italy; f.forastiere@deplazio.it; 4Institute for Biomedical Research and Innovation (IRIB), National Research Council (CNR), 90146 Palermo, Italy; 5Environmental Research Group, School of Public Health, Faculty of Medicine, Imperial College, London SW7 2AZ, UK; 6Department of Pediatrics, University of Alberta, Edmonton, AB T6G 2R3, Canada; 7Stollery Children’s Hospital, Edmonton, AB T6G 2R3, Canada; 8Anatomic Pathology Division, Department of Lab. Medicine and Pathology, Children’s Hospital of Eastern Ontario, Ottawa, ON K1H 8L1, Canada

**Keywords:** children, cancer, brain, epidemiology, carcinogenicity, diazinon, glyphosate, malathion, parathion, tetrachlorvinphos, International Association for the Research on Cancer, World Health Organization

## Abstract

Background: Brain tumors are the second most common neoplasm in the pediatric age. Pesticides may play an etiologic role, but literature results are conflicting. This review provides a systematic overview, meta-analysis, and IARC/WHO consideration of data on parental exposure to pesticides and childhood brain tumors. Methods: We searched PubMed, SCOPUS, and Google Scholar for literature (1 January 1966–31 December 2020) that assessed childhood brain tumors and parental exposure to pesticides. We undertook a meta-analysis addressing prenatal exposure, exposure after birth, occupational exposure, and residential exposure. A total of 130 case-control investigations involving 43,598 individuals (18,198 cases and 25,400 controls) were included. Results: Prenatal exposure is associated with childhood brain tumors (odds ratio, OR = 1.32; 95% CI: 1.17–1.49; I^2^ = 41.1%). The same occurs after birth exposure (OR = 1.22; 95% CI: 1.03–1.45, I^2^ = 72.3%) and residential exposure to pesticides (OR = 1.31; 95% CI: 1.11–1.54, I^2^ = 67.2%). Parental occupational exposure is only marginally associated with CBT (OR = 1.17, 95% CI: 0.99–1.38, I^2^ = 67.0%). Conclusions: There is an association between CBT and parental pesticides exposure before childbirth, after birth, and residential exposure. It is in line with the IARC Monograph evaluating the carcinogenicity of diazinon, glyphosate, malathion, parathion, and tetrachlorvinphos.

## 1. Introduction

Cancer is one of the most important causes of death in childhood in developed countries. In particular, childhood brain tumors (CBT) are the second most common pediatric cancer [1]. Despite the ongoing research, the etiology of this fatal tumor remains unknown. The most recent genomic and molecular analysis of the CBT’s biology, management, and prognosis has been impressive, with neuropathological classification linked to molecular biology results [2,3,4,5]. Most of the molecular subgroup-specific outcome data in CBT are generated from retrospective studies on heterogeneously treated children. It will be crucial to prospectively evaluate the clinical implications of genomic and molecular data in the context of therapeutic trials before the conventional clinical implementation.

Some of the factors that have been established include a few underlying genetic syndromes and ionizing radiation [6]. However, evidence regarding other potential risk factors, including pesticides exposure, carcinogen-metabolizing genes, and viruses, is limited and/or conflicting [7,8,9].

Several case-control studies on the effect of pesticide exposure on CBT have yielded mixed results [10,11,12]. In addition, the meta-analyses on the impact of pesticides on CBT have often recommended caution in interpreting results due to the possible implications of some potential confounding factors [13]. Hence, additional information reflecting a diversity of exposure situations will be more revealing to strengthen the existing findings.

A pesticide is a term used to describe a substance or mixture of substances essentially used to destroy, repel, or mitigate any pest. Commonly, pesticide use is considered more related to agriculture, and children living on or near farmlands are likely to be exposed through a number of mechanisms that include agro-industrial application drift, overspray, and dust brought inside homes on shoes of parents [14]. Another potential exposure involves urban settings when pesticides are used for lawn/garden care [15,16]. In addition, several studies have elucidated that various factors may increase the exposure to pesticides at home. These factors include inadequate or excessive application of pesticides and improper washing of pesticide-treated bedding. Therefore, children are likely to get exposed to pesticides, either breathing in or eating them [17,18,19,20,21,22,23]. Also, we examine the potential role of exposure to occupational pesticides, parental exposure to household pesticides, and children exposure before birth and after birth in CBT using a PRISMA-based systematic review and meta-analysis including one (case-control) study design. Our statements align with the IARC Monographs. Volume 112 of the monographs evaluated the carcinogenicity of diazinon, glyphosate, malathion, parathion, and tetrachlorvinphos [24,25].

## 2. Methods

### Study Selection, Criteria, and Data Extraction

A search on PubMed, Google Scholar, and Scopus was conducted between 1 January 1966 and 31 December 2020. Initially, an electronic search was done using the words pesticides OR herbicides OR fungicides OR insecticides and children OR childhood and childhood brain cancer, central nervous system (CNS) tumor, astrocytoma, primitive neuro-ectodermal tumor (PNET) tumors, and occupation, OR occupational with different combinations of the words pesticide (s), child, childhood brain tumor and parental occupation children, childhood, infant, newborn, preschool child, adolescent, youth, teenager, young adult, tumors, occupation, occupational, farmers, agriculture, horticulture, pesticide applicators, and residential household. Residential exposure location identifies an exposure location outside the confines of an industrial facility where individuals may reasonably be present for most hours of each day over many years. They include individual houses and areas that are zoned to allow residential use either exclusively or in conjunction with other services (https://www.lawinsider.com/dictionary/residential-exposure-location; accessed on 9 November 2021).

The next step was to check the reference lists of the publications to identify any additional studies. The search was limited to studies published in English in the open literature in peer-reviewed journals (Figure 1) according to the Preferred Reporting Items for Systematic reviews and Meta-Analyses (PRISMA) [26].

Inclusion criteria were articles with (1) individuals up to 24 years exposed to pesticides from parental occupational exposure (farmers/agricultural workers, pesticide applicators, workers engaged in the manufacture of pesticides and others such as horticulturists, greenhouse workers, gardeners, etc.), (2) targeting the link between childhood brain cancer and pesticides exposure, (3) without multiple studies, and (4) case-control design only. We excluded reviews, abstracts, letters, meta-analyses, case reports, pool analysis, and studies with the mixed population without an identifiable pediatric age group. Despite being potentially relevant, we did not include cohort studies because they have larger confidence intervals than case-control studies. Thus, they have reduced power in the case of rare diseases.

Since the estimated effect for the different study designs can be influenced, to varying degrees, by various sources of bias [26], only case-control studies were included in our meta-analysis. Given the rarity of the condition, the case-control study is the most suitable study design. We applied the Newcastle-Ottawa quality assessment scale to each study individually [27]. The quality assessment of studies included in the meta-analysis is vital for an evidence-based evaluation. Newcastle-Ottawa Scale (NOS), a quality assessment tool, is used worldwide for observational studies [28]. We regarded studies with a NOS score of 6 and above as good quality, while studies with a NOS score of less than six were considered poor quality. Analyzed data included the location of the study, number of participants, source of exposure, timing of exposure, and outcomes. Stratification was performed focusing on several variables that could influence the results, including variables related to exposure such as exposure time windows (prenatal, before conception, during pregnancy, and postnatal). Analysis was done using Comprehensive Meta-Analysis (CMA) application software [29]. The aim was to evaluate the association of CBT with pesticide exposure considering different exposure categories. We defined four groups to improve understanding of the potential involvement of household pesticide exposure and parental occupational exposure to pesticides in CBT. We separated (1) before birth, (2) after birth, (3) occupational, and (4) household exposures.

## 3. Statistics

After the studies and data were extracted, the categories were narrowed down to the most relevant variables for input into the CMA program [29]. These relevant variables included the sample size, OR, and CI (lower and upper limit). Since not all of the relevant variables were provided by the studies, some had to be calculated. First, the pooled values of the relevant variables were calculated, including the pooled variance-weighted least square mean with its variance and standard error [29,30]. Then, the 95% CI around the pooled effect size was obtained from the pooled variance. We opted for a mixed-effects analysis approach. The fixed-effects model pools effect sizes across the subgroups within each study, as we assumed variations due only to sampling errors. The random-effects model combined effect sizes across different studies, considering more variability due to between-study variance [31]. After computing effect sizes, we assessed the impact of heterogeneity and checked for publication bias. The I^2^ describes the percentage of total variation across studies that is due to heterogeneity rather than chance. I^2^ is computed as I^2^ = 100% × (Q − df)/Q, in which Q is Cochran’s heterogeneity statistic and df the degree of freedom [29,32,33]. F values lie between 0% and 100%, as values falling below zero are changed to zero. If I^2^ equals 0%, it indicates no observed heterogeneity, while a larger value shows rising heterogeneity.

We plotted the natural logarithm of the OR (lnOR) estimate versus the standard error (SE) to examine potential publication bias owing to study size. In addition, funnel plot asymmetry was assessed using Begg’s modified funnel plot and Egger’s regression asymmetry test [29,34].

## 4. Results

Table 1 shows the studies that were selected for the meta-analysis [1,6,7,8,9,15,16,35,36,37,38,39,40,41,42,43,44,45,46,47,48,49,50,51,52]. We examined the link between CBT and pesticide exposure before birth, including (Figure 2A) only case-control studies evaluating the pre-conception and pregnancy period. We found a pooled OR of 1.31 with a confidence interval of 1.17–1.46 and a *p*-value of <0.0001. There was moderate heterogeneity with I^2^ of 41.08. Figure 3A shows the funnel plot of standard errors vs. effect estimates for the exposure before the birth category, and no evidence of a cluster was found. The second subgroup (17 studies) involved studies on pesticide exposure after birth (Figure 2B). A pooled OR of 1.22 was found with a 95% confidence interval of 1.03–1.45 and a *p*-value of 0.021. Again, there was high heterogeneity as I^2^ statistic was 72.3%. The funnel plot of standard errors against effect size is shown in Figure 3B, and it does not indicate asymmetry. In the third instance, we explored whether occupational exposure was associated with CBT in a group of 20 studies. The result in Figure 2C shows that the effect of occupational exposure was small and not statistically significant. The pooled OR was 1.17 with a confidence interval of 0.99–1.38 and a *p*-value of 0.067. There was high heterogeneity between studies with the I^2^ of 67.1. Figure 3C shows the funnel plot of standard errors vs. effect estimates, and there was no evidence of asymmetry. The fourth exposure category (Figure 3D) involved studies on pesticide exposure through home/residence and whether there was an association with CBT. We arrived at a pooled OR of 1.31 with a confidence interval of 1.11–1.54 and a *p*-value of 0.001. Again, there was high heterogeneity between studies with an I^2^ of 67.2. Figure 3D shows the funnel plot of standard errors vs. effect estimates for the home residence category. No clear clustering pattern was identified. Additional testing for publication bias was undertaken for each exposure situation. Table 2 shows the summary of statistical procedures for assessing publication biases by pesticide exposure categories. The Egger’s test of the intercept yielded an insignificant *p*-value (Table 2), which was above 5%, regardless of the exposure category. It is essential to assess the impact of missing studies [53,54]. The resulting funnel plot contains only one imputed study on the left side of the graph in addition to the experimental studies. The adjusted point estimate (dark diamond) and the original analysis did not arrive strictly at the same odds ratio. With “Trim and Fill”, the imputed point estimate was 1.32 (CI = 1.17–1.49), suggesting a lower odds ratio than the original analysis. In addition, one small study (representing a large effect size) fell toward the right of the combined OR, while relatively none fell toward the left. It is a warning sign of minor possible publication bias. However, the adjusted point estimate was close to the original OR of 1.305. Therefore, if considering any such bias, its impact would still be modest. In other words, including all relevant studies might shift the effect size, but the key finding that parental exposure to pesticides before birth can cause CBT would probably remain unchanged. For the household exposure category, the Egger’s test was insignificant (Bo = 1.922; *p* = 0.082). Considering the eventual impact of any missing studies (Figure 3A), the adjusted point estimate (dark diamond) and the original analysis did not arrive strictly at the same odds ratio. The imputed point estimate was 1.248 (CI = 1.106–1.460), suggesting a lower odds ratio than the original analysis. The funnel plot is asymmetric, but the impact of the bias appears modest without changing the core findings.

## 5. Discussion

We found an association between CBT and parental exposure to pesticides, mainly when exposure to pesticides occurs before and after birth. In addition, our results showed that parental exposure to pesticides in household settings is more strongly associated with CBT than parental occupational exposure to pesticides. Farm residence can be used as a proxy measure for occupational and residential pesticide exposures, although it does not indicate the direct exposure of an individual [10]. In subgroup analysis, we grouped farm residence studies along with occupational exposure. It is paramount to remember that various potential carcinogens can be associated with farm life, and some have been extensively revised [55,56,57,58,59]. It is difficult to assess on several occasions if it is the exposure to animals, exposure to pesticides, or the combination of more than one factor that is more likely to be linked with CBT. For instance, pesticide-related to farms has been associated with cancer development, but farm animals and pets have been reported as a potential risk factor for CBT. Although the gut microbiome is influenced by the contacts with pets and may play a protective role, one substantial risk related to animal contact is getting exposed to animal viruses, such as coronaviruses. Since RNA viruses and DNA viruses have induced brain tumors and hematological and cardiac neoplasms [40,60], it may be plausible that COVID-19 may cause hyperactivation of immune cells, resulting in chronic inflammation, which may act as promoting factor. In some settings, it may induce the reawakening of dormant cancer cells such as neutrophil extracellular traps [61].

In this review and meta-analysis, cohort studies were not included. However, a previous meta-analysis [13] found an association between CBT and parental occupational exposure to pesticides in case-control studies and cohort investigations. The OR among case-control groups was 1.3; 95% CI: 1.11–1.53, and the summary risk ratio calculated included cohorts was 1.53; 95% CI: 1.20–1.95. The results in both cases show a similar OR. The difference could be attributed to the variations in exposure category definitions across studies. Additionally, in our review, studies in which parents’ exposure was limited to pesticide application in the home and/or garden only were included under the household/residential exposure category and not under the farming/agriculture profession. Finally, it is essential to note that another recent French population-based study that performed a pooled analysis also elucidated an association between childhood brain cancer and maternal use of household pesticides during pregnancy [62]. A limitation of our study is that there is a lack of information on specific pesticides. It would be great to study risks per type of pesticide or at least per group of pesticides, but it is challenging with the data available in the literature.

Our results suggest that the pest control treatments at home may be associated with the increased risk of CBT. However, looking at the individual studies, the residential pesticide appears to be different by the type of application. The method of application (professional extermination vs. non-professional application) also plays an essential role in the causation of CBT. Studies emphasizing pesticide application’s exact circumstances, including the types of personal protective equipment involved, may be important. The use of hand gloves at pesticide application, for instance, can be critical, as suggested by the Agricultural Health Study of the children of Iowa pesticide applicators in the USA. Although there was no association between parental exposure to pesticide application and the risk of childhood cancer in that study, parent applicators who did not use chemically resistant gloves ended up having children at higher risk of childhood cancer than those who used gloves [63]. Parental exposure to pesticides has been demonstrated to be strongly associated with this tumor. Previous meta-analyses have investigated a reasonably consistent link between pesticide exposure and CBT development [35,36,38,64]. Furthermore, the most significant risks of CBT appear to be associated with household insecticide use as well as prenatal exposure to insecticides [35]. In addition, it was also suggested that genetic susceptibilities might have a role in determining the effects of childhood pesticide exposures, and thus recommended future studies to examine the role of gene-environment interactions in the development of CBT [65]. Also, it was investigated that children, when exposed to insecticides, were more likely to develop brain tumors if they also carried *PON1_C-108T_* SNP (single nucleotide polymorphism of the paraoxonase—*PON1* gene) [10]. Constitutive, genetic variation influences insecticide metabolism, and Searles Nielsen et al. [46] also examined whether CBT is associated with this SNP. They found no association between CBT and the single coding region SNP *PON1_Q192R_* but a strong dose–response relationship between CBT and *PON1_C–108T_*, a promoter-region SNP associated with enzyme levels [46]. These authors also reported that genotype and insecticide interactions occur during childhood but usually not during pregnancy. In fact, it seems that during prenatal development, maternal enzymes serve as the first line of defense against exogenous exposure. Thus, even though they do not suggest a lack of effect of insecticide exposure during this potentially sensitive period, they emphasize the lack of synergism with fetal genotype.

In a population-based case-control study done between 1978–1990, the purpose of which was to determine if CBT is associated with some functional genetic polymorphisms, the researchers found no biologically plausible main effects for any of the metabolic polymorphisms with CBT risk. They noticed strong interactions between genotype and insecticide exposure during childhood. These interactions were present among both Hispanic and non-Hispanic white children. Similar results were also observed with two other gene variants, *FMO1_C-9536A_* and *BCHE_A539T_*. The latter is thought to affect the ability to detoxify organophosphorus and/or carbamate insecticides [39]. Hence, it further emphasizes that genetic susceptibility is an essential factor in the exposure of pesticides and CBT development.

It is crucial to target the studies’ plausibility, strength, consistency, specificity, and temporality. In the studies considered, the authors argued that pesticides are taken to be biologically active molecules. Therefore, it is entirely plausible that they contribute to the etiology of cancer since this hypothesis does not conflict with the present understanding of the natural history of cancer. Along the same lines, every case-control study and every cohort study compare the disease rates in exposed and non-exposed individuals and, thus, fulfill the criterion of experiment listed by Hill [66]. The strength of the CBT and pesticide relationship can be considered high, as reflected by the relative risk in the studies and the confidence intervals.

Moreover, we also know that genetic susceptibility seems to be a key player in the casual relationship of pesticide exposure and CBT, making the association even more robust. As for the specificity, it is a factor that does appear to impact the results. Most of the studies often used broader pesticides: herbicides, insecticides, fungicides, or other types of pesticides. However, the studies which gave good details of type of pesticide, timing, and frequency seem to have reported an increased association between pesticide exposure and CBT. As for temporality, we can say that exposure before birth is prior to the diagnosis of CBT. Most CBTs are initiated in embryonal development. Thus, at least exposure after birth will occur too late as the first cause of cancer.

Nevertheless, according to the multi-hit theory, we might also expect later exposure to affect the course of the disease, tumor growth, and tumor fate.

In terms of consistency, the results of meta-analysis and studies do not show much variation. Hence, it can be inferred that the relationship between CBT and parental/childhood pesticide exposure is consistent.

Considering our findings, global and concerted actions to raise awareness of CBT risk and pesticide exposure are worthwhile. For instance, experts from countries already met at the international level to assess the carcinogenicity of the organophosphate pesticides, which include tetrachlorvinphos, parathion, malathion, diazinon, and glyphosate [25,67,68,69,70,71,72]. In the future, it will be more helpful to focus on parental exposure and pediatric cancer while emphasizing evidence-based, proactive measures such as the wearing of chemical-resistant gloves during pesticide application.

Due to the participation of two co-authors (FF, CS) to IARC/WHO monographs on pesticides, this systematic review and meta-analysis have taken a relatively more encompassing view than previous ones to examine in depth the effects of exposure to pesticides on CBT under different time windows of exposure from early human development stages (e.g., before birth, during pregnancy, and after birth) and in socio-economic setting attributes (e.g., professional and residential exposures). In addition, a unique feature of this review is that residential exposure to pesticides was restricted to data from the literature about parental use of pesticides only in the home and garden. The IARC Monographs identify human cancer’s preventable causes (e.g., chemicals, physical and biological agents, pharmaceuticals, complex mixtures, and occupational exposures). Volume 112 provides evaluations of the carcinogenicity of some organophosphate insecticides and herbicides, including diazinon, glyphosate, malathion, parathion, and tetrachlorvinphos [55,56,57,58,59]. Awareness and education programs regarding the adverse effects of pesticide exposure on health may be beneficial to minimize the exposure, particularly in a time of repeated drought and climate change [73]. Warmer temperatures enhance evaporation, decreasing surface water and drying out soils and vegetation, which can be heavily loaded with chemicals.

In conclusion, the results of our meta-analysis support an association between pesticide exposure and CBT. If it occurs before or during pregnancy and in early childhood, the pesticides exposure is more likely to contribute to developing cancer of young/adult brain tumors. Additionally, our study strongly supports that pesticide exposure via household plays a role in CBT development. Because of the etiologic complexity of brain tumors, it is highly recommended that future studies be extensive, and investigations with transgenic animal models may be useful. Moreover, enhanced methods for exposure assessment, time window of the stage of human development, and pesticide application methods will be required.

Additionally, our studies suggest after synthesis that awareness and education programs regarding the harmful effects of pesticide exposure on health may benefit the population. Residential use of pesticides should be carried out with all precautions to minimize exposure. Until more extensive studies are performed to confirm the results, all efforts should be maximized to limit children and adults’ exposure to pesticides. We strongly advocate the continuous and thorough perusal of the IARC/WHO Monographs on pesticides and insecticides by chemical industries, food agencies, and consumers regulators.

## Figures and Tables

**Figure 1 children-08-01096-f001:**
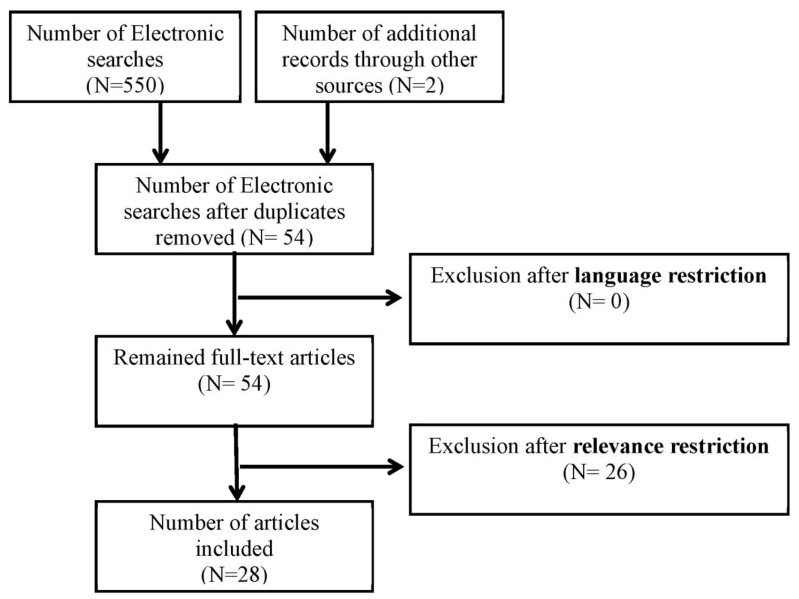
PRISMA Steps followed to select articles for the study.

**Figure 2 children-08-01096-f002:**
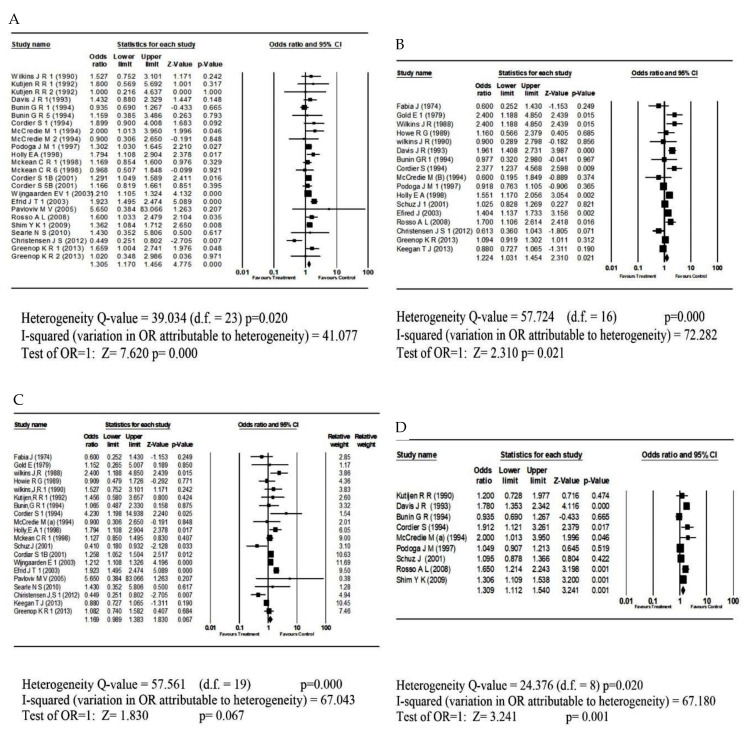
Forest plot displaying a meta-analysis of the effect of pesticides on Child Brain: (**A**) (upper Left) = Before birth; (**B**) (upper right) = After Birth; (**C**) (lower left) =Occupational Exposure; (**D**) (lower right) = Residential Exposure.

**Figure 3 children-08-01096-f003:**
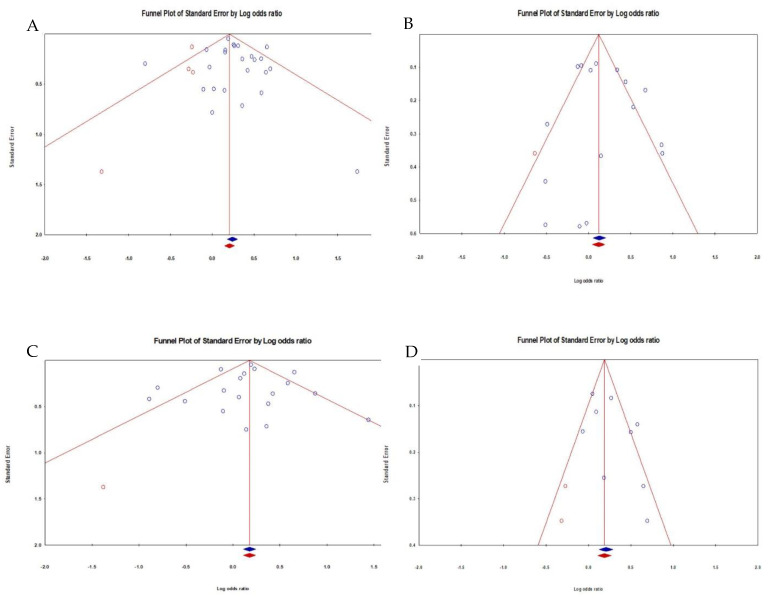
Funnel Plot with imputed studies - Effects of household pesticides on CBT. (**A**) (upper Left) = Before birth; (**B**) (upper right) = After Birth; (**C**) (lower left) = Occupational Exposure; (**D**) (lower right) = Residential.

**Table 1 children-08-01096-t001:** Characteristics of studies included in the meta-analysis.

Study	Source and Time of Exposure	Effect Size	LL	UL	Total Cases	Total Controls	No of Cases Events	No of Control Events
Fabia [1]	Farm Exposure of Father at the time of childbirth birth	0.6	0.25	1.42	386	772	6	78
Gold [6]	Extermination for insects	1.2			84	73, 78 *^1^	14	12
Gold [6]	Child living on farm	1			84	73, 78 *^2^	9	9
Wilkins [7]	Father’s occupation at the time of birth	2.4	1.2	4.9	491	30	20	
Howe [8]	Contact with herbicides, insecticides	0.94 ^Ψ^	0.47	1.9	74	138	19	38
Wilkins [9]	Postnatal	0.9	0.3	2.9	110	193	4	N/A
Wilkins [9]	Prenatal	1.6	0.4	6.1	110	193	4	N/A
Wilkins [9]	Preconception	2.7	0.8	9.1	110	193	6	N/A
Kuijten [16]	Exposure to Insecticides	1.2	0.7	1.9	163	163	38 *^3^	33
Kuijten [15]	Preconception	1.8	0.6	6	163	163	11 *^4^	6
Kuijten [15]	Pregnancy	1	0.2	4.3	163	163	5 *^5^	5
Kuijten [15]	Postnatal	1.3	0.7	6.3	163	163	5 *^6^	4
Davis [35]	Pesticide used for nuisance pest during pregnancy	1.8	0.8	4	45	85, 108 *^7^	30	15
Davis [35]	Pesticide used for nuisance pest during pregnancy	1.8	0.8	4	45	85, 108 *^8^	30	15
Davis [35]	Pesticide use for nuisance pest from birth to 6 months	1.9	0.8	4.3	45	85, 108 *^9^	28	16
Davis [35]	Pesticide used for nuisance pest from 7 months to diagnosis	3.4	1.1	10.6	45	85, 108 *^10^	38	6
Davis [35]	Pesticide used for termite from 7 months of age till diagnosis	1.4	0.5	3.9	45	85, 108 *^11^	12	33
Davis [35]	Pesticide used for lice from 7 months of age till diagnosis	1.3	0.4	4.1	45	85, 108 *^12^	8	37
Davis [35]	Kwell used for lice from 7 months of age till diagnosis	4.6	1	21.3	45	85, 108 *^13^	7	37
Davis [35]	Insecticide used in during pregnancy	1.5	0.6	3.9	45	85, 108 *^14^	11	34
Davis [35]	Insecticide used in garden or orchard from birth to 6 months of age	2.3	0.7	8.3	45	85, 108 *^15^	7	37
Davis [35]	Insecticide used in garden or orchard from 7 months of age to diagnosis	1.6	0.7	3.6	45	85, 108 *^16^	22	22
Davis [35]	Herbicide used in garden or orchard during pregnancy	1.1	0.5	2.5	45	85, 108 *^17^	12	33
Davis [35]	Herbicide used in garden or orchard from birth to 6 months of age	1.7	0.7	3.9	45	85, 108 *^18^	15	29
Davis [35]	Herbicide used in garden or orchard from 7 months of age till diagnosis	2.4	1	5.7	45	85, 108 *^19^	30	34
Bunin [36]	Home pest extermination	0.7	0.4	1.4	322	268	24	31
Bunin [36]	Home pest extermination	1	0.6	1.9	322	268	34	33
Bunin [36]	Insect spray or pesticides	1.5	0.8	2.7	322	268	34	26
Bunin [36]	Insecticide spray or pesticides	0.7	0.4	14	322	268	31	39
Bunin [36]	Farm residence of mother during entire pregnancy	0.5	0.1	1.8	322	268	5	8
Bunin [36]	Farm residence of mother during entire pregnancy	3.7	0.8	23.9	322	268	14	6
Bunin [36]	Child living on Farm for more than a year	0.4	0.1	1.6	322	268	6	9
Bunin [36]	Child living on Farm for more than a year	5	1.1	46.8	322	268	14	6
Cordier [37]	Farm residence of mother during pregnancy	2.5	0.4	16.1	75	109	4	2
Cordier [37]	Farm residence of child in childhood	6.7	1.2	38	75	109	8	2
McCredie [38]	Mother lived/worked on a farm month before or during pregnancy	0.9	0.3	2.6	82	164	5	11
McCredie [38]	Child exposure via lived/worked on a farm	0.6	0.2	1.9	82	164	4	12
Pogoda [10]	Termite, prenatal exposure	2.7	0.5	14.2	224	218	5	2
Podoga [10]	Nuisance pest prenatal	1.1	0.8	1.7	224	218	106	97
Podoga [10]	Insecticides, prenatal exposure	1.3	0.7	2.4	224	218	26	20
Podoga [10]	Herbicides, prenatal exposure	0.9	0.1	6.1	224	218	2	3
Podoga [10]	Snail killer, prenatal exposure	1.1	0.6	2.1	224	218	21	18
Podoga [10]	Flea/tick, prenatal exposure	1.7	1.1	2.6	224	218	76	53
Podoga [10]	Termite childhood exposure	0.7	0.4	1.3	224	218	23	32
Podoga [10]	Nuisance pest childhood exposure	1	0.6	1.5	224	218	150	146
Podoga [10]	Lice, childhood exposure	0.6	0.4	1	224	218	38	50
Podoga [10]	Insecticide, childhood exposure	1.2	0.8	2	224	218	57	47
Podoga [10]	Herbicide, childhood exposure	1.2	0.3	4.9	224	218	4	4
Podoga [10]	Fungicide, childhood exposure	0.1	0	1	224	218	1	8
Podoga [10]	Snail killer childhood exposure	1	0.6	1.8	224	218	41	38
Podoga [10]	Flea/tick childhood exposure	1	0.7	1.4	224	218	106	102
Holly [39]	Child was on a farm for more than a year	1.7	0.88	3.1	540	801	21	19
Holly [39]	Child was on a farm for less than a year	1.2	0.58	2.6	540	801	13	16
Holly [39]	Child less than 6 months of age when first time on farm	1.9	0.96	3.8	540	801	19	15
Holly [39]	Child more than 6 months of age when on farm	1.5	0.59	2.4	540	801	15	19
Holly [39]	Child ever lived/worked on farm before reference date	1.5	0.9	2.4	540	801	35	36
Holly [39]	Maternal exposure to agriculture pesticides	1.8	0.77	4.2	540	801	12	10
Holly [39]	Mother lived/worked on a farm for one or more months before or during pregnancy	1.6	0.86	2.9	540	801	22	21
Holly [39]	Mother livestock farm employment 5 year preceding the index childbirth	7.4	0.86	64	540	801	5	1
Mckean-Cowdin [40]	Father exposure 5 years before birth	1.2	0.65	2.1	540	801	22	27
Mckean-Cowdin [40]	Father exposure 5 years before birth	1.3	0.65	2.6	540	801	13	27
Mckean-Cowdin [40]	Father exposure 5 years before birth	0.99	0.33	2.9	540	801	4	27
Mckean-Cowdin [40]	Father exposure 4 years preconception	1.2	0.65	2.1	540	801	21	26
Mckean-Cowdin [40]	Father exposure during pregnancy	0.99	0.43	2.3	540	801	10	14
Mckean-Cowdin [40]	Mother exposure 5 years before birth	0.93	0.34	2.6	540	801	6	11
Mckean-Cowdin [40]	Mother exposure 5 years before birth	1.3	4.3	3.8	540	801	5	11
Mckean-Cowdin [40]	Mother exposure 4 years preconception	0.87	0.29	2.6	540	801	5	10
Mckean-Cowdin [40]	Mother exposure during pregnancy	1.2	0.33	4.4	540	801	4	6
Schüz [41]	Pesticide use on farms	0.41	0.18	0.93	466	2458	7	84
Schüz [41]	Pesticide use on farms	0.41	0.18	0.93	466	2458	7	84
Schüz [41]	Household use of pesticide more than 1/year	1.19	0.81	1.77	466	2458	38	164
Schüz [41]	Pesticide use in garden	0.94	0.68	1.29	466	2458	60	290
Cordier [42]	Father exposed to pesticides 5 years before birth	1.3	1	1.8	1218	2223	80	104
Cordier [42]	Father exposed to pesticides 5 years before birth	1.2	0.8	1.8	1218	2223	N/A	N/A
Cordier [42]	Father exposed to pesticides 5 years before birth	1.1	0.6	1.9	1218	2223	N/A	N/A
Cordier [42]	Father exposed to pesticides 5 years before birth	1.8	1	3.5	1218	2223	N/A	N/A
Cordier [42]	Mother exposed to pesticides 5 years before birth	1.1	0.7	1.9	1218	2223	22	39
Cordier [42]	Mother exposed to pesticides 5 years before birth	1.2	0.6	2.3	1218	2223	N/A	N/A
Efird [43]	Child on a farm	1.3	1	1.7	1218	2223	92	164
Efrid [43]	Agriculture job	1.9	1.3	2.8	1218	2223	56	55
Efrid [43]	Agriculture job	1.8	1.1	3	1218	2223	30	32
Efrid [43]	Agriculture job	2	1.2	3.2	1218	2223	35	33
Efrid [43]	Child on a farm first when less than 6 months of age	1.6	1.1	2.2	1218	2223	62	89
Efrid [43]	Agriculture job	2.4	0.79	7.1	1218	2223	7	6
Van Wijingaarden [44]	Parental occupation	1.212	1.108	1.326	20**			
Pavloviv [45]	Father exposure to occupational pesticides, preconception	5.65	0.61	131.85	60	60	4	0
Searles Nelson [46]	Parental exposure to organophosphates before birth	1.38	0.56	1.38	66	236	9	14
Rosso [47]	Father engaged in lawn care during pregnancy	1.6	1	2.4	318	318	105	67
Rosso [47]	Father engaged in lawn care after childbirth	1.7	1.1	2.6	318	318	111	70
Shim [48]	Insecticide ever used in garden/lawn in a 2-year period before birth	1.3	0.9	2	526	526	52	39
Shim [48]	Insecticide used in garden/lawn in 2-year period before childbirth	1.1	0.6	1.9	526	526	25	23
Shim [48]	Herbicide used in garden/lawn in 2-year period before childbirth	1.9	1.2	3	526	526	53	27
Shim [48]	Herbicide used in garden/lawn in 2-year period before childbirth	1	0.6	1.8	526	526	26	24
Shim [48]	Fungicide used in garden/lawn in 2-year period before childbirth	1.8	0.7	4.6	526	526	13	7
Shim [48]	Fungicide used in garden/lawn in 2-year period before childbirth	1.5	0.3	3.8	526	526	8	6
Shim [48]	Insecticide used in garden/lawn in 2-year period before childbirth by father	1	0.6	1.8	526	526	28	26
Shim [48]	Insecticide used in garden/lawn in 2-year period before childbirth by father	1	0.5	1.9	526	526	20	19
Shim [48]	Herbicide used in garden/lawn in 2-year period before childbirth by father	2	1.2	3.4	526	526	40	20
Shim [48]	Herbicide used in garden/lawn in 2-year period before childbirth by father	1.1	0.5	2	526	526	21	18
Shim [48]	Fungicide used in garden/lawn in 2-year period before childbirth by father	3.1	0.3	30	526	526	3	1
Shim [48]	Fungicide used in garden/lawn in 2-year period before childbirth by father	3.6	0.4	32.6	526	526	4	1
Shim [48]	Insecticide used in GARDEN/LAWN in 2-year period before childbirth by mother	1	0.5	1.9	526	526	18	18
Shim [48]	Insecticide used in GARDEN/LAWN in 2-year period before childbirth by mother	0.7	0.3	2.2	526	526	6	8
Shim [48]	Herbicide used in garden/lawn in 2-year period before childbirth by mother	1.9	0.7	4.8	526	526	13	7
Shim [48]	Herbicide used in garden/lawn in 2-year period before childbirth by mother	0.8	0.3	2.5	526	526	6	7
Shim [48]	Fungicide used in garden/lawn in 2-year period before childbirth by mother	1.7	0.6	4.8	526	526	10	6
Shim [48]	Fungicide used in garden/lawn in 2-year period before childbirth by mother	1.6	0.4	6.9	526	526	5	3
Searle Nielsen [49]	Mother exposure to insecticide one month preconception until birth	1.43	0.35	5.77	201	286	4	4
Christensen [50]	Postnatal farm exposure	0.57	0.28	1.17	352	646	12	43
Christensen [50]	Postnatal farm exposure	0.57	0.21	1.53	352	646	6	43
Christensen [50]	Postnatal farm exposure	0.89	0.24	3.34	352	646	4	43
Christensen [50]	Farm residence (primary residence of mother during pregnancy)	0.38	0.12	1.16	352	646	4	38
Christensen [50]	farm residence (primary residence of mother during pregnancy)	0.86	0.21	3.55	352	646	3	38
Christensen [50]	Farm residence (primary residence of mother during pregnancy)	0.4	0.19	0.88	352	646	9	38
Greenop [51]	Any pest treatment during this period	0.84	0.56	1.26	335	1363	44	165
Greenop [51]	Any termite treatment	1	0.59	1.72	335	1363	22	78
Greenop [51]	Any general treatment for spider/insects	1.03	0.73	1.43	335	1363	80	260
Greenop [51]	Child home during treatment	1.63	1.02	2.6	335	1363	38	91
Greenop [51]	Child home during treatment	1.63	1.02	2.6	335	1363	38	
Greenop [51]	Child slept in room within 3 days of treatment	0.88	0.57	1.34	335	1363	45	169
Greenop [51]	Paternal exposure any time before pregnancy	1.07	0.68	1.68	335	1363	32	103
Greenop [51]	Paternal exposure year before pregnancy	1.11	0.55	2.23	335	1363	13	27
Keegan [52]	Paternal occupation, agriculture	0.88	0.73	1.07	11,874	11,874	228	251

Notes: *^1^ = 73 normal controls 78 cancer control; *^2^ = 73 normal controls 78 cancer control; *^3^, *^4^, *^5^ = Ratio of case exposed to control unexposed and case unexposed and control exposed pairs; From *^6^ to *^19^ = 85 Friend control 108 Cancer controls; 20** = Summarized the data; ^Ψ^ Risk Ratio; N/A = Not Available; LL, Lower Limit; UL, Upper Limit; No of Case Events, Number of cancer events in the cases group; No of Control Events, Number of cancer events in the control group.

**Table 2 children-08-01096-t002:** Summary results of the statistical procedures for assessing publication biases by pesticide exposure categories.

Pesticides Exposure	N	Summary Effect Size	Egger’s Test		Duval and Tweedie’s Test **	
			B_0_	*p*-Value	Point Estimate (No Imputation)	Imputed Estimate
Before birth	24	1.32 *	0,29	0.257	1.315	1.302
After Birth	17	1.22 *	0.78	0.189	1.224	1.186
Occupational	20	1.17	−0.04	0.476	1.169	1.162
Household	8	1.31 *	1.92	0.082	1.309	1.248

* Significant at less than 5%. N is the number of studies; ** Test performed under a random effects model; B_0_ is the intercept.

## Data Availability

Data is contained within the article.

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
