# Peer review of "Parental Pesticide Exposure and Childhood Brain Cancer: A Systematic Review and Meta-Analysis Confirming the IARC/WHO Monographs on Some Organophosphate Insecticides and Herbicides"

_children, 2021, doi:10.3390/children8121096_

Round 1

Reviewer 1 Report

In this study, the authors made a systematic overview, meta-analysis, and IARC/WHO consideration of data on parental exposure to pesticides and childhood brain tumors. They concluded that there is an association between childhood brain tumors and parental pesticides exposure before childbirth, after birth, and residential exposure. The manuscript is straightforward, well written, and concise and has clear results within the scope of a retrospective analysis.  It's a valuable contribution to the “children” journal. Some minor flaws need to be addressed

Minor points:

[1] “Introduction”, Page 2/22, Lines 40-41:

Despite the ongoing research, the etiology of this fatal tumor remains unknown.”.

At that point, please make a comment about the genomic and molecular analysis that has allowed for significant advances in our understanding of the biology, management and prognosis of pediatric brain tumors. However, the majority of the molecular subgroup-specific outcomes data in pediatric brain tumors are generated from retrospective analyses on heterogeneously treated patients. Consequently, it is important to prospectively evaluate the clinical implications of genomic and molecular data in the context of therapeutic trials before the standard clinical implementation.

[2] “Discussion”, Page 17/22, Lines 313-317:

Constitutive genetic variation influences insecticide metabolism and Searles Nielsen et al. [42] also examined whether CBT is associated with this SNP. They found no association between CBT and the single coding region SNP PON1Q192R but a strong dose–response relationship between CBT and PON1C– 108T, a promoter-region SNP associated with enzyme levels [42].”.

They also reported that the presence of interactions between genotype and insecticide exposure occurs during childhood, but generally not during pregnancy. During prenatal development, maternal enzymes serve as a first line of defense against exogenous exposures. Also, perhaps fetal expression of some enzymes is too low, regardless of genotype, to alter insecticide dose sufficiently to protect the brain; here again, maternal enzymes may be important. Data of Searles Nielsen et al do not suggest a lack of effect of insecticide exposure during this potentially sensitive period, but rather a lack of synergism with fetal genotype.

Author Response

In this study, the authors made a systematic overview, meta-analysis, and IARC/WHO consideration of data on parental exposure to pesticides and childhood brain tumors. They concluded that there is an association between childhood brain tumors and parental pesticides exposure before childbirth, after birth, and residential exposure. The manuscript is straightforward, well written, and concise and has clear results within the scope of a retrospective analysis. It's a valuable contribution to the “children” journal. Some minor flaws need to be addressed

Many thanks for your comments. We do appreciate the valuable input in peer reviewing our manuscript.

Minor points:

[1] “Introduction”, Page 2/22, Lines 40-41:

“Despite the ongoing research, the etiology of this fatal tumor remains unknown.”.

At that point, please make a comment about the genomic and molecular analysis that has allowed for significant advances in our understanding of the biology, management and prognosis of pediatric brain tumors. However, the majority of the molecular subgroupspecific outcomes data in pediatric brain tumors are generated from retrospective analyses on heterogeneously treated patients. Consequently, it is important to prospectively evaluate the clinical implications of genomic and molecular data in the context of therapeutic trials before the standard clinical implementation.

We extended this paragraph introducing four very recent (2021) references.

[2] “Discussion”, Page 17/22, Lines 313-317:

“Constitutive genetic variation influences insecticide metabolism and Searles Nielsen et al. [42] also examined whether CBT is associated with this SNP. They found no association between CBT and the single coding region SNP PON1Q192R but a strong dose–response relationship between CBT and PON1C– 108T, a promoter-region SNP associated with enzyme levels [42].”.

They also reported that the presence of interactions between genotype and insecticide exposure occurs during childhood, but generally not during pregnancy. During prenatal development, maternal enzymes serve as a first line of defense against exogenous exposures. Also, perhaps fetal expression of some enzymes is too low, regardless of genotype, to alter insecticide dose sufficiently to protect the brain; here again, maternal enzymes may be important. Data of Searles Nielsen et al do not suggest a lack of effect of insecticide exposure during this potentially sensitive period, but rather a lack of synergism with fetal genotype

We extended this paragraph introducing your precious comments and suggestions.

Reviewer 2 Report

I applaud the authors for dealing with such an important topic. In general, I find, the meta-analysis is well conducted. Of course, it would be more interesting to study risks per type of pesticide or at least per group of pesticides. But I do understand that this is difficult. At least the authors should acknowledge the lack of information on specific pesticides as a weakness of their study.

I do find the mentioning of COVID-19 in this context unnecessary (paragraph starting at line 60)! I do not agree with the authors that social distancing measures will per se lead to an increased exposure to pesticides (as the authors again propose later, in the discussion). But here it is certainly not the place as the pandemic, even if it might have an impact on pesticide exposure, will not affect currently or previously conducted case-control studies.

Inclusion criteria, starting at line 118: mentions only occupational exposure. But the authors also examine residential exposure. By the way, “residential exposure” needs to be better defined. Only later do they discuss residential exposures at farm houses. But they also mention urban use of pesticides. And certainly there are residential applications of pesticides (in fighting ants, cockroaches, mosquitoes, etc.).

Inclusion criteria, continued. Only later do the authors explain why they did not include cohort studies. That explanation should be provided here. And I am sorry that they did not include cohort studies. I agree that cohort studies have reduced power in the case of rare diseases. But still, cohort studies exist. And even if they find larger confidence intervals, their point estimates would support the findings from the case-control studies.

Line 154: please reconsider the sentence “In the little former,…” The reader is really hard pressed to understand what you mean by “the little former”!

Table 1: What is the meaning of “LL”, “UL” and of “No of Cases Events” and “No of Control Events”? I do assume that LL and UL are lower and upper limit (of the 95 % confidence interval?) of the “effect size”. And the effect size, I assume is measured as OR? I do understand number of cases. But what are “case events”?

Table 1 continued: This is a very long table. Many studies are reported repeatedly as they are used in multiple models. In total, the table therefore is very difficult to read. Footnotes are only found many pages later. The authors ran models on “before birth”, “after birth”, “occupational” and “household” (for which I am not sure: also occupational or household exposures can either occur before or after birth, or, also, both before and after birth). But if it really was possible to discern these exposure scenarios: Why not group the studies according to these scenarios and turn table 1 into 4 tables?

The second sentence of the “results” section states that the first subgroup consisted of 18 studies. These studies are certainly not the first 18 studies from table 1!

Figure 2: the pictures and especially the letters of the text are rather small. Is it necessary to lump 4 figures together into one (and, above all, into 2 columns)? If the authors wanted to reduce the number of figures, I would rather skip figure 3, as figure 4 provides the same information plus some extra information.

Line 276ff: “The results in both cases differ from ours could be attributed to an error in reporting occupational exposure by parents between treatment and control groups within each study.” First, the text is not good English. But more to the point, I am not sure what they wanted to say. Both meta-analyses found a significantly increased odds ratio. The odds ratio in the other study was larger. So what? They examined different studies. But what is the meaning of “treatment group”? Who was treated? Or do the authors want to say that case-control studies are prone to reporting bias? Yes, but would this lead to an underestimation of effect (they only included case control studies, the other meta-analysis also cohort studies)? As far as I understand, when the other study only included case-control studies, they found an OR very similar to the present study. And generally, in the following discussion it gets increasingly clear that in the methods section the reader was not sufficiently well informed about the definition of household exposure and occupational exposure.

The whole discussion on causal interpretation starting in line 286 is a bit confusing. Yes, it seems the association is causal. But no, we cannot safely say so, more research is needed. Then the authors start discussing single studies (Iowa pesticide applicators). I believe this was a cohort study, but I am not sure. Did the authors include that study in their meta-analysis? As also shown in the forest plots, null or even negative associations do occur in single studies. This is why we perform meta-analyses!

I do not understand the discussion of the so-called Bradford-Hill Criteria. (By the way, one reference, the one from 1965, should suffice. You need not cite the 50th anniversary re-print also!) But certainly, both cohort and case-control studies are observational and not experimental studies. On the other hand, why do the authors state (line 343) that “As for temporality, we cannot confidently say a relationship between pesticide exposure and CBT development.”? Of course, exposure before birth is prior to the diagnosis of CBT! Yes, there is the theory that most CBT are initiated in embryonal development. In that case, at least exposures after birth will come too late as the first cause of cancer. But according to the multi-hit theory, we might expect also later exposures to affect the course of the disease, tumor growth and tumor fate.

Minor:

In the abstract the authors introduced an acronym (CBT). This only makes sense if that acronym is then also used throughout the abstract. But in line 25 again the full name is written.

Figure 1: needs to be improved. Boxes are too small to read all letters.

Author Response

I applaud the authors for dealing with such an important topic. In general, I find, the meta-analysis is well conducted. Of course, it would be more interesting to study risks per type of pesticide or at least per group of pesticides. But I do understand that this is difficult. At least the authors should acknowledge the lack of information on specific pesticides as a weakness of their study. I do find the mentioning of COVID-19 in this context unnecessary (paragraph starting at line 60)! I do not agree with the authors that social distancing measures will per se lead to an increased exposure to pesticides (as the authors again propose later, in the discussion). But here it is certainly not the place as the pandemic, even if it might have an impact on pesticide exposure, will not affect currently or previously conducted case-control studies.

Many thanks for your comments. We do appreciate the valuable input in peer reviewing our manuscript.

Inclusion criteria, starting at line 118: mentions only occupational exposure. But the authors also examine residential exposure. By the way, “residential exposure” needs to be better defined. Only later do they discuss residential exposures at farm houses. But they also mention urban use of pesticides. And certainly there are residential applications of pesticides (in fighting ants, cockroaches, mosquitoes, etc.).

We specifically defined "residential exposure". Thank you for pointing it out.

Inclusion criteria, continued. Only later do the authors explain why they did not include cohort studies. That explanation should be provided here. And I am sorry that they did not include cohort studies. I agree that cohort studies have reduced power in the case of rare diseases. But still, cohort studies exist. And even if they find larger confidence intervals, their point estimates would support the findings from the case-control studies.

We added a limitation to the study in not having investigated and included cohort studies.

Line 154: please reconsider the sentence “In the little former,…” The reader is really hard pressed to understand what you mean by “the little former”!

We deleted this sentence, which was misleading. Thank you!

Table 1: What is the meaning of “LL”, “UL” and of “No of Cases Events” and “No of Control Events”? I do assume that LL and UL are lower and upper limit (of the 95 % confidence interval?) of the “effect size”. And the effect size, I assume is measured as OR? I do understand number of cases. But what are “case events”?

We added the notes to each table portion to make it clear. Thank you!

Table 1 continued: This is a very long table. Many studies are reported repeatedly as they are used in multiple models. In total, the table therefore is very difficult to read. Footnotes are only found many pages later. The authors ran models on “before birth”, “after birth”, “occupational” and “household” (for which I am not sure: also occupational or household exposures can either occur before or after birth, or, also, both before and after birth). But if it really was possible to discern these exposure scenarios: Why not group the studies according to these scenarios and turn table 1 into 4 tables?

We were discussing a long time if we had to split table 1 in several tables, but we went to the conclusion that one unique table is useful because all studies are grouped, despite being long. Our Statistician and epidemiologist, Dr. Feulefac, was suggesting it properly. Thus, we humbly ask you to understand the reasons of our choice. Thank you!

The second sentence of the “results” section states that the first subgroup consisted of 18 studies. These studies are certainly not the first 18 studies from table 1!

We modified the text, which was misleading. Thank you!

Figure 2: the pictures and especially the letters of the text are rather small. Is it necessary to lump 4 figures together into one (and, above all, into 2 columns)? If the authors wanted to reduce the number of figures, I would rather skip figure 3, as figure 4 provides the same information plus some extra information.

Thank you for your suggestion. We deleted figure 3, because as you correctly said, it provides the same information as figure 4, which is now figure 3. Thank you!

Line 276ff: “The results in both cases differ from ours could be attributed to an error in reporting occupational exposure by parents between treatment and control groups within each study.” text is not good English. But more to the point, I am not sure what they wanted to say. Both meta-analyses found a significantly increased odds ratio. The odds ratio in the other study was larger. So what? They examined different studies. But what is the meaning of “treatment group”? Who was treated? Or do the authors want to say that case-control studies are prone to reporting bias? Yes, but would this lead to an underestimation of effect (they only included case control studies, the other meta-analysis also cohort studies)? As far as I understand, when the other study only included case-control studies, they found an OR very similar to the present study. And generally, in the following discussion it gets increasingly clear that in the methods section the reader was not sufficiently well informed about the definition of household exposure and occupational exposure.

We clarified the definitions and the text and thank you for the elucidation on our study. Very much appreciated!

The whole discussion on causal interpretation starting in line 286 is a bit confusing. Yes, it seems the association is causal. But no, we cannot safely say so, more research is needed. Then the authors start discussing single studies (Iowa pesticide applicators). I believe this was a cohort study, but I am not sure. Did the authors include that study in their meta-analysis? As also shown in the forest plots, null or even negative associations do occur in single studies. This is why we perform meta-analyses!

We included it indeed and we clarified the causal interpretation. Many thanks for your help.

I do not understand the discussion of the so-called Bradford-Hill Criteria. (By the way, one reference, the one from 1965, should suffice. You need not cite the 50 anniversary re-print also!) But certainly, both cohort and case-control studies are observational and not experimental studies. On the other hand, why do the authors state (line 343) that “As for temporality, we cannot confidently say a relationship between pesticide exposure and CBT development.”? Of course, exposure before birth is prior to the diagnosis of CBT! Yes, there is the theory that most CBT are initiated in embryonal development. In that case, at least exposures after birth will come too late as the first cause of cancer. But according to the multi-hit theory, we might expect also later exposures to affect the course of the disease, tumor growth and tumor fate.

We extended the text introducing the multi-hit theory and deleted one of the two references (so-called Bradford-Hill Criteria). Thank you!

Minor: In the abstract the authors introduced an acronym (CBT). This only makes sense if that acronym is then also used throughout the abstract. But in line 25 again the full name is written.

The abbreviation was deleted in the abstract. Thank you!

Figure 1: needs to be improved. Boxes are too small to read all letters.

We changed figure 1 and increased the font size of all letters in the boxes. Thank you!
